# Efficacy and Safety of Micropulse Transscleral Cyclophotocoagulation

**DOI:** 10.3390/jcm11123447

**Published:** 2022-06-15

**Authors:** Victor. A. de Vries, Jan Pals, Huub J. Poelman, Parinaz Rostamzad, Roger C. W. Wolfs, Wishal D. Ramdas

**Affiliations:** Department of Ophthalmology, Erasmus Medical Center, 3000 CA Rotterdam, The Netherlands; v.a.devries@erasmusmc.nl (V.A.d.V.); j.pals@erasmusmc.nl (J.P.); h.poelman@erasmusmc.nl (H.J.P.); p.rostamzad@erasmusmc.nl (P.R.); r.wolfs@erasmusmc.nl (R.C.W.W.)

**Keywords:** glaucoma, intraocular pressure, IOP-lowering medication, micropulse, trans-scleral cyclophotocoagulation

## Abstract

Background: Early studies have shown that micropulse transscleral cyclophotocoagulation (MP-TSCPC) might be an effective and safe treatment option for lowering intraocular pressure (IOP). These studies were, however, somewhat limited, in particular by their retrospective nature and the length of follow-up. Therefore, we assessed the efficacy and safety of this novel treatment in a large cohort for up to 4 years. Methods: We performed a prospective cohort study, including all patients who were treated with MP-TSCPC since November 2017. The primary outcome was a reduction of IOP and the number of IOP-lowering medications. Results: The mean ± standard deviation baseline IOP and number of IOP-lowering medications were 26.6 ± 10.8 mmHg and 3.3 ± 1.3. IOP was reduced by 8.2 ± 7.9 (31.8% reduction), 6.9 ± 8.7 (28.1% reduction), and 7.1 ± 8.4 (30.2% reduction) mmHg after 6, 12, and 24 months, respectively (*p* < 0.001). The mean postoperative number of IOP-lowering medications was significantly reduced after 6 months by 0.6 ± 1.5 (*p* = 0.002) but was not significantly different after 12 or 24 months. Oral acetazolamide was significantly reduced from 28 (29%) eyes before treatment, to 9 (9%) at the last follow-up visit (*p* < 0.001). No major complications were observed after treatment. Conclusions: MP-TSCPC is a safe and effective treatment option for lowering IOP, but only reduced IOP-lowering medications in the first 6 months after treatment. However, MP-TSCPC is especially effective in getting patients off oral IOP-lowering drugs.

## 1. Introduction

Glaucoma is a neurodegenerative eye disease that has been the most common cause of irreversible blindness worldwide for several decades. At present, only one modifiable risk factor has been identified: (increased) intraocular pressure (IOP). The optic nerve can be damaged by either continuously high IOP or rapid increases thereof, which in turn can cause irreversible visual field loss [1]. Currently, the only effective treatment for glaucoma has been decreasing IOP by either reducing the production of aqueous humor, or increasing the aqueous outflow [2].

Over the years several treatment modalities have been established to accomplish this goal, including medical treatment, laser treatment, and surgery [3,4]. As a general principle, the first step consists of topical medication and/or laser treatment. Surgical intervention is usually reserved for a later stage and typically used to treat either refractive glaucoma or to reduce the need for continuous topical medication. Traditionally, the two most commonly used surgical treatments have been the trabeculectomy and the implantation of a glaucoma drainage device. Additionally, minimally invasive glaucoma surgery (MIGS) has been developed, with the aim of achieving a reduction in IOP with a better safety profile than conventional surgery [5].

As a last resort for patients with uncontrolled, refractory glaucoma, cyclodestructive procedures can be used, including cyclocryocoagulation, cyclodiathermy, and trans-scleral cyclophotocoagulation [6]. These treatments can cause serious complications such as persistent hypotony, persistent intraocular inflammation, hyphema, decreased visual acuity, and phthisis bulbi [6,7,8]. As a result, many ophthalmologists do not perform these procedures altogether or only use them when all other glaucoma treatments have failed.

Recently, the micropulse transscleral cyclophotocoagulation (MP-TSCPC) has been developed with the aim of damaging less of the ciliary body compared to conventional cyclodestructive techniques [9]. This is achieved by applying laser energy to the ocular tissue only for short bursts followed by rest periods. Theoretically, this cycling mode allows for the accumulated energy in the target pigmented epithelium to reach the critical threshold for photocoagulation, while giving the non-target tissues (e.g., the nonpigmented epithelium and ciliary body stroma) enough time in between pulses to dissipate the accrued energy to the surrounding tissues. Early studies have shown that MP-TSCPC can reduce IOP in patients with refractory glaucoma, with substantially less severe complications compared to conventional cyclodestructive treatments [10,11,12,13]. Consequently, the place of MP-TSCPC in the glaucoma treatment paradigm is being reconsidered. Unfortunately, the current evidence for the efficacy and safety of MP-TSCPC is still limited, in particular, because of small study populations and short follow-up periods. Therefore, we performed a prospective cohort study including all consecutive patients who were treated with MP-TSCPC, to assess its efficacy and safety.

## 2. Materials and Methods

### 2.1. Study Population

This study is part of the Erasmus Glaucoma Cohort Study, an ongoing prospective study started in 2017. It includes all patients with glaucoma that visited the outpatient clinic of the Ophthalmology Department of the Erasmus Medical Centre, Rotterdam, The Netherlands, since January 2015 (that is, partially retrospective) [14]. For the purpose of this study, we included all patients who underwent MP-TSCPC between November 2017 and June 2021. To maximize the generalizability of the results, we did not exclude patients based on indication or diagnosis (e.g., primary or secondary glaucoma).

Baseline characteristics included age, sex, ethnicity, preoperative IOP, number of IOP-lowering medications, untreated IOP at time of diagnosis, preoperative best-corrected visual acuity (BCVA), refraction, visual field mean deviation (MD) and pattern standard deviation (PSD), pachymetry, medical history, and family history for glaucoma. The Medical Ethics Committee of the Erasmus University has approved the study. Formal consent was not required because patients did not undergo non-clinically related interventions.

### 2.2. Procedure

All patients were treated following a standardized treatment protocol. All treatments were performed by or under the supervision of two glaucoma surgeons (WDR and RCWW) at the operation theatre. After administering subtenon anesthesia, patients underwent MP-TSCPC with a first-generation MicroPulse P3 probe of the Iridex cyclo G6 laser system (Iris Medical Instruments, Mountain View, CA, USA) at 2000 mW for a duration of 80 s (or 90 s if preoperative IOP was >30 mmHg) per hemisphere using 10 s sweeps at a 31.3% duty cycle. During retreatments, power was increased up to 2500 mW for up to 90 s per hemisphere. The MP3 probe was applied in a continuous movement with steady pressure on the ocular globe. Previous sites of glaucoma surgery (site of drainage implant or blebs), places of scleral thinning, and the 3 and 9 o’clock hours were avoided. Eye ointment containing hydrocortisone, oxytetracycline, and polymyxine B (Terra-Cortril^®^) was applied immediately following treatment. Dexamethasone 0.1% eye drops were prescribed thrice daily, which were tapered off over the course of 9 days. Postoperatively, IOP-lowering medications were withdrawn if appropriate.

### 2.3. Assessment of Outcomes

The IOP was measured using Goldmann applanation tonometry (Haag-Streit, Köniz, Switzerland). The number of IOP-lowering medications was calculated by adding the number of different categories of medication. The categories were: beta-blockers, prostaglandin analogues, alfa2-agonists, carbonic anhydrase inhibitors, and oral acetazolamide. Fixed combinations of eye drops were calculated as two separate drugs [15]. Baseline characteristics were collected at the last visit before treatment. Follow-up data was collected for postoperative visits at day 1, 1 week, 1 month, 3 months, 6 months, 12 months, 18 months, 24 months, and their last recorded visit. The primary outcome was a reduction of IOP and the number of IOP-lowering medications. Secondary outcomes were the cumulative incidence of treatment failure, defined as an IOP out of the target range (<20% reduction compared to baseline IOP) for two consecutive visits after at least one month of follow-up, and the cumulative incidence of requiring secondary treatment (including non-topical treatments other than MP-TSCPC). If an eye required a secondary treatment during follow-up, IOP measurements were only recorded until the date of retreatment to minimize confounding. Hypotony was defined as an IOP < 5 mmHg at two or more consecutive visits (excluding the measurement one-day after treatment). The postoperative BCVA and visual field MD and PSD was recorded at the earliest after 3 months or for the first measurement thereafter.

### 2.4. Statistical Analyses

Linear mixed models with repeated measurements were applied to assess differences in IOP and IOP-lowering medications over time. Two models were created in which one of these variables was fitted as the dependent variable with visits as a (fixed) factor, assuming an unstructured correlation matrix. The models were adjusted for age and gender (both fixed effects). The confidence intervals were adjusted for multiple comparisons using the Bonferroni method. As the model for IOP-lowering medications could not reach convergence, its results were considered unreliable. Therefore, we also analyzed differences between pre- and postoperative IOP and the number of IOP-lowering medications with paired t-tests. For patients who received retreatments, differences in outcomes between primary treatments and retreatments were analyzed using independent t-tests. To make the results of this study comparable to other studies, we also analyzed our data according to the recommendations of the World Glaucoma Association [16]. We performed Kaplan–Meier analyses for treatment failure (defined as <20% IOP reduction compared to baseline for 2 consecutive visits after one month of follow-up) and the requirement of secondary treatment, as well as a scatter plot for both primary and secondary treatments. To assess predictors for secondary treatment, we used independent t-tests for continuous variables and chi-square tests (or Fisher’s exact tests where applicable) for categorical variables to analyze differences in baseline characteristics between outcome groups. All attributes with a univariate *p*-value < 0.1 were subsequently fitted into a multivariate logistics regression model to calculate odds ratio’s with corresponding 95% confidence intervals (CI) for risk of treatment failure or the requirement of secondary treatment. The threshold for statistical significance was identified as a *p*-value of 0.05 or less. All statistical analyses were performed using SPSS v28.0.1.0 for Windows (IBM Inc., Chicago, IL, USA).

## 3. Results

A total of 96 eyes of 84 patients underwent 114 treatments (Table 1). The median follow-up was 1.0 year, with an interquartile range of 0.5–1.9 (total range of 0.0–3.8) years. The mean ± standard deviation visual acuity was 0.4 ± 0.4 Snellen, and the visual acuity was lower than 0.05 Snellen (>1.3 LogMar) in 25 (26%) of eyes. Seven eyes (8%) had no light perception at all. The mean ± standard deviation visual field MD was −19.31 ± 11.33, and the mean ± standard deviation visual field PSD was 8.00 ± 3.21. One patient died during follow-up after 15 months.

Figure 1 presents the mean IOP and the number of IOP-lowering medications over time. The mean ± standard deviation preoperative IOP was 26.6 ± 10.8 mmHg. The linear mixed model showed that the mean ± standard deviation postoperative IOP was significantly reduced (*p* < 0.001) at all measured follow-up periods to 20.2 ± 8.8, 16.6 ± 7.2, 18.2 ± 8.5, 18.6 ± 9.2, 17.6 ± 6.9, 17.3 ± 6.3, 15.7 ± 7.5, 17.0 ± 6.5, and 20.3 ± 11.1 mmHg at day 1, 1 week, 1 month, 3, 6, 12, 18, 24 months, and at the last follow-up visit, respectively. The linear mixed model and paired t-tests showed markedly similar results for IOP over time (Appendix A). The mean ± standard deviation preoperative number of IOP-lowering medications was 3.2 ± 1.2. Paired t-tests showed that the mean postoperative number of IOP-lowering medications was significantly reduced to 2.7 ± 1.4, 2.6 ± 1.2, 2.6 ± 1.3, and 2.6 ± 1.4 at 1 month, 3 months, 6 months postoperatively and at the last follow-up visit, respectively. Oral acetazolamide (Diamox^®^) was used for 28 (29%) eyes before treatment, which could be reduced to 9 (9%) eyes at the last follow-up visit (*p* < 0.001).

Figure 2 shows the relationship between pre- and postoperative IOP for individual eyes, and those with a 20% reduction in IOP. An IOP reduction of at least 20% was achieved in 65% (47/72), 62% (28/45), 65% (16/23) and 58% (66/113) of eyes after 6 months, 1 year, 2 years, and at last follow-up visit, respectively (Figure 3A). Concerning predictors associated with a higher risk of treatment failure, an increased preoperative IOP was significantly associated with a decreased likelihood of treatment failure. The risk of treatment failure had an odds ratio of (95% CI): 0.93 (0.88—0.98) per mmHg increase in preoperative IOP (Appendix A). Neither age, sex, ethnicity, visual field MD, diagnosis, spherical equivalent, previous ocular surgery, treating surgeon, nor a family history of glaucoma were significantly associated with treatment failure (Appendix A). A total of 36 (32%) eyes required secondary treatment (Figure 3B), with a mean ± standard deviation follow-up of 7.3 ± 6.3 months and 84% of treatments occurring within the first year of follow-up. Of all eyes that required secondary treatment (N = 36), 18 (50%) underwent a second MP-TSCPC treatment, 17 (47%) underwent glaucoma surgery, and 1 (3%) underwent cryocoagulation. Indications for secondary treatment were persistent high IOP (N = 33), desire for further reduction in medication (N = 2), and increased loss of visual field despite adequate IOP reduction (N = 1). Contrary to the treatment failure rate, baseline IOP was not significantly associated with the requirement of secondary treatment, nor were any other baseline characteristics (Appendix A). For those eyes which received a second MP-TSCPC treatment, pretreatment IOP and IOP-lowering medication did not differ significantly between the initial treatment and the retreatment. Similarly, the mean IOP and IOP-lowering medication was not significantly different for any of the posttreatment follow-up points. After receiving a second MP-TSCPC treatment, 5 (29%) eyes required a third intervention after mean ± standard deviation 5.0 ± 3.9 months. Of these, 2 were MP-TSCPC treatments, 1 was a XEN-implant, 1 was an Ahmed implant, and 1 was a cryocoagulation. An IOP reduction of at least 20% was achieved for 13 (77%) of these eyes at the last follow-up visit.

No major complications, including hypotony, were observed after treatment. One patient developed a hyphema one year after MP-TSCPC treatment, however, this was attributed to rubeosis after central retinal vein occlusion. Additionally, one patient with end-stage glaucoma and no light-perception underwent an evisceration after 10 months because of sustained pain and irritation. Baseline mean visual field MD was −19.38 ± 11.23 dB, which decreased marginally to −19.44 ± 8.7 dB after treatment (*n* = 27; *p* = 0.044). Baseline visual field PSD was 8.19 ± 3.64 dB, which did not significantly differ after treatment (*p* = 0.62) at 8.53 ± 3.13 dB. Furthermore, no statistically significant difference was found between pre- and posttreatment visual acuity at 0.44 ± 0.42 and 0.42 ± 0.43 (*p* = 0.12), respectively. Postoperative BCVA was reduced by >1 line for 17 (17%) eyes.

## 4. Discussion

### 4.1. Summary of Findings

In this study, we examined the efficacy of MP-TSCPC in a diverse cohort of patients with a broad range of stages and etiologies of glaucoma. The mean IOP was consistently reduced by 7–8 mmHg or 30–35% up to 2 years after treatment. An IOP reduction of at least 20% was achieved in approximately 65% of patients. The number of IOP-lowering medications was reduced in the period of 1 to 6 months after treatment, but returned to preoperative levels after 1 year. Secondary treatment was required in 32% of eyes during follow-up, most commonly another MP-TSCPC procedure. No complications which could be directly linked to the MP-TSCPC treatment were reported.

### 4.2. Relationship with Literature

The treatment parameters used in this study were most comparable to those of Zaarour (N = 75) and De Crom et al. (N = 141, Appendix A) [13,17]. The results of De Crom et al. were in line with our results, with a mean reduction in IOP of 7–8 mmHg or 30% up to 2 years after treatment. Their success rate was also similar to our results at 60%. Zaarour et al. found that the IOP was reduced significantly by 9–10 mmHg or 40% up to 15 months after treatment. Their success rate (IOP between 6 and 21 mmHg or >20% IOP reduction) of 81.4% at 6 months and 73.6% at 12 months was somewhat higher than ours. Considering the observational nature of these studies, it is difficult to retrospectively determine the cause of this difference. Interestingly, our study and that of De Crom et al. included mainly Caucasian patients. MP-TSCPC might be more effective in study populations with a different ethnic composition. Our analyses did not show a correlation between ethnicity and the rate of treatment failure or the requirement of secondary treatment, but our study might have been underpowered to detect such a relationship. Another reason could be that Zaarour et al. applied the probe for a slightly longer duration (180 [2 *×* 90 s] vs. 160 [2 × 80 s] s), as previously published literature has shown that both efficacy and complication rate correlates with the total amount of energy employed [18].

Nguyen et al. (N = 95) found a 30% reduction in IOP after one year, a result comparable to our findings [19]. They did, however, find a relatively high success rate (>20% IOP reduction) of 77%. There were some differences between this study and ours. Most notably, whereas we used a power of 2000 mW during all primary treatments, increasing this only sporadically to 2100–2500 mW during retreatments, Nguyen et al. used 2000–2500 mW for 2 × 90 s during primary treatments and up to 3000 mW for secondary MP-TSCPC procedures. This might, in part, also explain their higher complication rate, with 11% of patients developing a keratopathy and 6% developing hyphema. Secondly, there were some differences in the study populations. Contrary to our study, Nguyen et al. (N = 95) included mostly patients at a relatively early stage of glaucoma. They also included a large number of patients (25%) with pseudoexfoliation and no patients with neovascular glaucoma or glaucoma secondary to uveitis. Considering the markedly higher success rate combined with a similar mean reduction in IOP, it is possible that MP-TSCPC provides a more consistent result for patients at an earlier stage of disease progression. Similar to Nguyen et al., Sarrafpour et al. (N = 73) found a higher average IOP reduction of 10 mmHg or 46% [20]. They did, however, find an IOP reduction of only 30% in patients who received treatment at 2000 mW, aligning with our findings, while observing a 57% IOP reduction in patients treated at 2500 mW. These findings correspond with the theory that MP-TSCPC follows a dose–response pattern related to the power used.

Our cumulative incidence of secondary treatments was 32%, which was slightly higher than the 21% of Nguyen et al. but comparable to the 30% of eyes reported by De Crom et al., once again reaffirming the difference in outcomes based on procedure parameters. Whether or not a second treatment is necessary can usually be determined within the first year of follow-up. After a second MP-TSCPC treatment, the IOP and the number of IOP-lowering medications were not significantly different for all follow-up visits compared to those same follow-up visits after the first treatment. However, 71% of these patients did not require a third treatment. The modest efficacy of a second MP-TSCPC treatment should be weighed against other treatment options on a case-by-case basis if the initial treatment proves insufficient. Oral acetazolamide usage was reduced by 68%, which is consistent with the overall trend in the existing literature of approximately 70% reduction after MP-TSCPC treatment (Appendix A).

### 4.3. Strength and Weaknesses

One of the strengths of this study is its relatively long follow-up of up to 2 years, and as the Erasmus Glaucoma Cohort Study is still ongoing, a longer follow-up with a larger cohort will be available in the future. Secondly, all MP-TSCPC procedures were performed by the same two surgeons using the same technique each time following a standardized protocol. Both surgeons were also always involved in determining the indication for treatment. Thirdly, the indication for MP-TSCPC gradually changed over the course of this study. Initially, almost all patients included had advanced to end-stage glaucoma. As the surgeons became more familiar with the procedure and observed its efficacy with no major complications, the indication for this treatment was progressively broadened.

Nevertheless, some limitations have to be considered. Most notably, due to the observational nature of this study, some unknown confounding or predisposing factors may have been missed. A second limitation was our moderate sample size with incomplete data during follow-up. As mentioned previously, the Erasmus Medical Center is a tertiary clinic. As a result, many patients were referred back to their peripheral hospital or clinic for further follow-up, and only part of that follow-up data was retrieved. In addition, to include as many patients as possible, a lower follow-up frequency was sometimes tolerated, especially for patients with long travel times. Furthermore, due to our relatively high rate of secondary treatments, follow-ups were often cut short. These combined factors produced small sample sizes for our later follow-up intervals, although the reduction in IOP remained consistent. Thirdly, two different surgeons performed the treatment. There were no significant differences in IOP or IOP-lowering medications between both surgeons, except for baseline IOP (29.32 ± 11.01 vs. 23.85 ± 9.35 mmHg; *p* = 0.008) and IOP-lowering medications 1 month after treatment (mean ± standard deviation 3.13 ± 1.45 vs. 2.31 ± 1.20; *p* = 0.019). Furthermore, there was no significant difference in any of the other IOP or IOP-lowering medication results, nor in the rates of treatment failure (*p* = 0.133) or secondary treatment (*p* = 0.794). Another weakness of this study was that we could not use both the “complete success” and “qualified success” endpoints as described in the WGA guidelines [16]. The disparity between these two outcomes depends on the usage of IOP-lowering medications, and almost all of our patients used at least some IOP-lowering medications both before and after treatment. Finally, we were not able to use a repeated measurements model for IOP-lowering medications.

### 4.4. Role in Glaucoma Practice

Any novel glaucoma treatment must find its place within the existing treatment paradigm. Currently, the most effective topical treatments, such as topical prostaglandins and beta-blockers, have been shown to lower IOP by 28–33% [21]. The XEN-implant, one of the most common minimally invasive glaucoma surgery (MIGS) procedures, has been shown to reduce IOP reduction by approximately 29% [14]. Our study shows that MP-TSCPC reduced IOP by 30–35%. Considering the fact that this is relatively low compared to other studies on MP-TSCPC, its efficacy appears to be comparable, if not superior, to topical therapy and MIGS. It does, however, remain inferior to the more traditional Baerveldt and Ahmed implants, which have been shown to reduce IOP by 43% [22].

The safety profile of MP-TSCPC seems, at least according to our data, to be superior to that of conventional glaucoma surgery and perhaps even MIGS. It should be noted that some studies have reported much higher complication rates for MP-TSCPC [18], but their treatment protocols were noticeably different than ours. Nevertheless, we should be particularly careful with the implementation of MP-TSCPC in common practice considering its cyclodestructive nature. Unlike glaucoma implant surgery, which is at least in theory reversible, the damage done to the ciliary body by MP-TSCPC can only partially regenerate [23]. The quintessential study design for examining the efficacy of a treatment is the RCT, none of which have yet been published on MP-TSCPC. The efficacy and low complication rate we found with our treatment protocol suggest that MP-TSCPC could be a relatively non-invasive option for lowering IOP, especially for patients at high risk for incisional surgery or for whom incisional surgery has previously proven ineffective.

## 5. Conclusions

In summary, MP-TSCPC is a safe and effective treatment option for lowering IOP. However, reduction of IOP-lowering medications was only maintained in the first 6 months after treatment. Especially for patients who are at high risk for incisional surgery or for whom incisional surgery has previously proven ineffective, MP-TSCPC could be an useful alternative. Randomized clinical trials are required to further analyze its efficacy and to minimize the chance confounding or predisposing factors are missed.

## Figures and Tables

**Figure 1 jcm-11-03447-f001:**
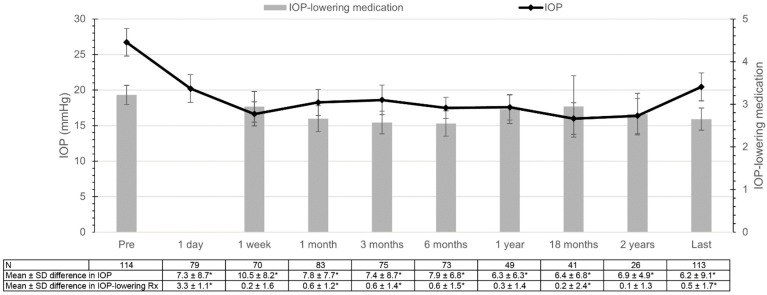
Mean intraocular pressure (IOP; line) and number of IOP-lowering medications (bars) with corresponding 95% confidence intervals, using a linear mixed model for IOP and paired t-tests for IOP-lowering medications. The “last” time interval had a median (interquartile range) follow-up of 11.99 (5.8—22.9) months. * = *p* < 0.05; IOP = intraocular pressure; SD = standard deviation; Rx = medications.

**Figure 2 jcm-11-03447-f002:**
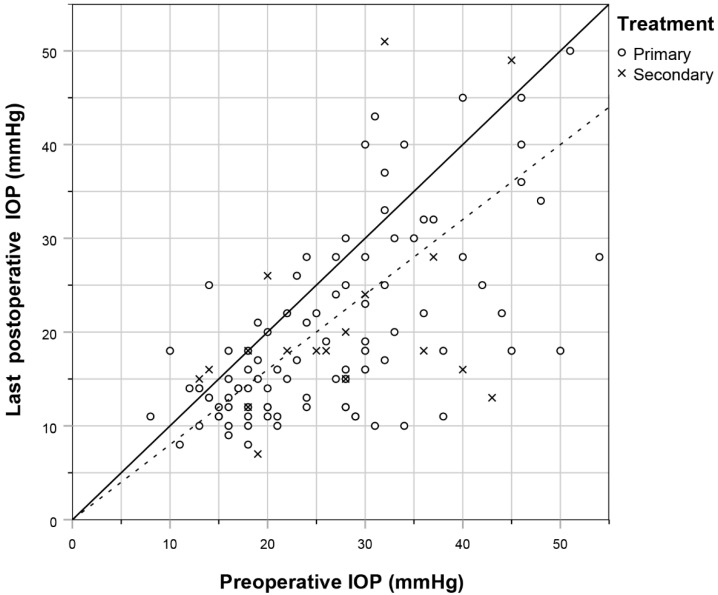
Scatterplot showing the pre- and postoperative IOP of primary (circles) and secondary (crosses) MP-TSCPC treatments. The dotted line represents an IOP reduction of 20%.

**Figure 3 jcm-11-03447-f003:**
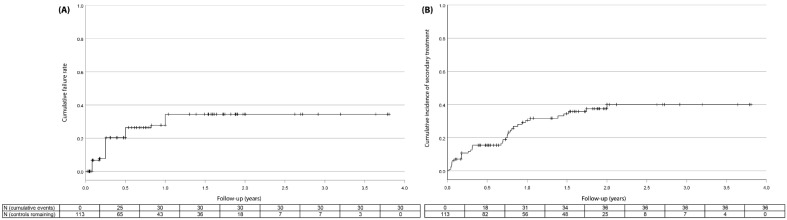
Kaplan–Meier cumulative incidence curve for failure defined as <20% reduction in IOP (**A**) and for secondary treatment (**B**). Censored patients are denoted by vertical tick marks. The Tables represent the total cumulative incidence.

**Table 1 jcm-11-03447-t001:** Baseline preoperative characteristics, presented as mean ± standard deviation unless stated otherwise.

Eyes (N, Patients)	96 (84)
Treatments (N, secondary treatments)	114 (18)
Age (years)	65.7 ± 14.0
Sex, female (N, %)	42 (44)
Caucasian descent (N, %)	90 (79)
Glaucoma etiology, (N, %)	
Primary	51 (53)
Neovascular	13 (14)
Uveitis	14 (15)
Trauma	2 (2)
Postvitrectomy	3 (3)
Complicated phaco procedure	2 (2)
Pseudoexfoliation syndrome (PEX)	1 (1)
Pigment dispersion syndrome (PDS)	2 (2)
Other	9 (8)
Untreated IOP at time of diagnosis (mmHg)	27.5 ± 12.2
Baseline IOP (mmHg)	26.6 ± 10.8
Number of IOP-lowering medications	3.4 ± 1.3
Visual acuity (LogMar)	0.4 ± 0.4
Spherical equivalent (D) *	−1.7 ± 2.5
Central corneal thickness (µm)	528 ± 47
Visual field mean deviation (MD; dB)	−19.38 ± 11.23
Visual field pattern standard deviation (PSD; dB)	8.00 ± 3.21
Positive family history (N, %)	24 (25)
Prior glaucoma surgery (N, %) **	26 (27)

* = before cataract surgery if applicable; ** = 11 trabeculectomies, 8 Baerveldt glaucoma implants, 5 Ahmed glaucoma implants, 6 XEN gel-stent implants, and 1 InnFocus microshunt implant; IOP = Intraocular pressure.

## Data Availability

The data presented in this study are available on request from the corresponding author.

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
