# Peer review of "Efficacy and Safety of Micropulse Transscleral Cyclophotocoagulation"

_jcm, 2022, doi:10.3390/jcm11123447_

Round 1
Reviewer 1 Report
de Vries et al. investigated the efficacy and safety of MP-TSCPC. I have some comments.
This was a prospective cohort study. The number of preoperative eyes was 114. However, the number of eyes were 79 and 70 at day 1 and 1 week, respectively. Why were so many data lacked?
The definition of treatment failure was an IOP out of target range and the requiring secondary treatment. 30 eyes were treatment failure and 36 eyes required secondary treatment. The number of eyes were correct?
IOP was significantly differ between no secondary treatment required and secondary treatment required in Supplemental table S2. However, baseline IOP was not significantly associated with requirement of secondary treatment (Lines 178-179).
The authors should show the table of multivariate logistic regression.
Author Response
Thank you for helping us improve our manuscript with your feedback.
Please see the attachment for our point-by-point response.

Reviewer 2 Report
The authors present a prospective study assessing the efficacy and safety of micropulse CPC for the treatment of glaucoma. They report a mean IOP reduction of 7.1 mm Hg after 24 months, a reduction in number of IOP-lowering medications by 0.6 that was significant only at 6 months, and a failure rate of 35% at 2 year follow up. Given the lack of large prospective studies involving MPCPC, there is a need for long-term studies to establish the efficacy and safety of the procedure. The authors conclude that MPCPC is a safe and efficacious treatment option for lowering IOP, though, there are a significant number of patients lost to follow up (almost 75%).
Major issues
- What was the time used for a single sweep of the hemisphere? The total amount of time and power for each hemisphere is reported, but there are no details regarding the sweep speed. This contributes to fluency values that have been reported to have affect efficacy.
- Which generation of probe was used? Given the length of the study, were generation 1 and generation 2 probes used? If so, this could contribute to effectiveness given that the power delivered depends on the generation of probe.
- Paired t-tests were used for IOP comparisons. A more correct test would be a repeated measures test as repeated measures were made over time.
- What is total number of N in the study? The text reports 96 eyes but Figure 1 shows an N of 114.
- What was the mean follow-up time for the last visit in Figure 1? There is a significant number of patients lost to follow up by 2 years. Therefore, it is informative to know when many of these values were recorded.
- Criteria for failure should include complete and qualified success which includes the use of medications according to WGA criteria. Failure should also include secondary treatment or need for additional glaucoma surgery
- What was the criteria for censoring in Figure 3? Need to include N’s at each time point.
Minor issues
- What were the other glaucoma in Table 1?
Author Response

(The authors gave the same response as above.)

Round 2
Reviewer 1 Report
I have no comments.